# VARIANCE REDUCTION IN HIERARCHICAL VARIATIONAL AUTOENCODERS

## ABSTRACT

Variational autoencoders with deep hierarchies of stochastic layers have been known to suffer from the problem of posterior collapse, where the top layers fall back to the prior and become independent of input. We suggest that the hierarchical VAE objective explicitly includes the variance of the function parameterizing the mean and variance of the latent Gaussian distribution which itself is often a high variance function. Building on this we generalize VAE neural networks by incorporating a smoothing parameter motivated by Gaussian analysis to reduce higher frequency components and consequently the variance in parameterizing functions and show that this can help to solve the problem of posterior collapse. We further show that under such smoothing the VAE loss exhibits a phase transition, where the top layer KL divergence sharply drops to zero at a critical value of the smoothing parameter that is similar for the same model across datasets. We validate the phenomenon across model configurations and datasets.

## 1 INTRODUCTION

Variational autoencoders (VAE) [10] are a popular latent variable model for unsupervised learning that simplifies learning by the introduction of a learned approximate posterior. Given data $x$ and latent variables $z$, we specify the conditional distribution $p(x|z)$ by parameterizing the distribution parameters by a neural network. Since it is difficult to learn such a model directly, another conditional distribution $q(z|x)$ is introduced to approximate the posterior distribution. During learning the goal is to maximize the evidence lower bound (ELBO), which lower bounds the log likelihood, $\log p(x) \geq \mathbb{E}_{q(z|x)} \big[ \log p(x|z) + \log p(z) - \log q(z|x) \big]$. In their simplest form, the generative model $p(x|z)$ and the approximate posterior $q(z|x)$ are Gaussian distributions optimized in unison.

A natural way to increase the modeling capacity of VAE is to incorporate a hierarchy of stochastic variables. Such models, however, turn out to be difficult to train and higher levels in the hierarchy tend to remain independent of input data – a problem termed *posterior collapse*. Posterior collapse in VAEs manifests itself by the latent distribution tending to fall back to the prior. With hierarchical VAEs the effect is found to be more pronounced in the top layers farther from the output. For the purpose of the paper and for clarity of exposition, we focus on the simplest extension of hierarchical variational autoencoders where stochastic layers are stacked serially on top of each other [2, 21] , $p(x, z) = p(x|z_1)p(z_L) \prod_{i=1}^{L-1} p(z_i|z_{i+1})$ and $q(z|x) = q(z_1|x) \prod_{i=1}^{L-1} q(z_{i+1}|z_i)$. The intermediate distributions in this model are commonly taken to be Gaussian distributions parameterized by neural network functions, so that $p(z_i|z_{i+1}) = \mathcal{N}(z_i|\mu(z_{i+1}), \sigma(z_{i+1}))$, where $\mu(z), \sigma(z)$ are neural networks computing the mean and variance of the Gaussian distribution. We refer to them as *vanilla hierarchical variational autoencoders*. For each stochastic layer in this model there is a corresponding KL divergence term in the objective given by

$$\mathbb{E}[KL(q(z_i|z_{i-1})||p(z_i|z_{i+1}))]. \tag{1}$$

As described later, expression 1 can be easily decomposed to show an explicit dependence on the *variance* of the parameterizing functions $\mu(z_i), \sigma(z_i)$ of the intermediate Gaussian distribution. We further show the KL divergence term to be closely related to the harmonics of the parameterizing function. For complex parameterizing functions the KL divergence term has large high frequency components (and thus high variance) which leads to unstable training causing posterior collapse.

Building on this, we suggest a method for training the simplest hierarchical extension of VAE that avoids the problem of posterior collapse without introducing further architectural complexity [13, 21]. Given a hierarchical variational autoencoder, our training method incorporates a smoothing parameter (we denote this by $\rho$) in the neural network functions used to parameterize the intermediate latent distributions. The smoothing is done such that expected values are preserved, the higher frequencies are attenuated and the variance is reduced. Next, the gradients computed with the smooth functions are used to train the original hierarchical variational autoencoder.

For the construction of the smoothing transformations for VAEs with Gaussian latent spaces we make use of ideas from the analysis of Gaussian spaces. We analyze the stochastic functions in vanilla hierarchical VAEs as Hermite expansions on Gaussian spaces [9]. The Ornstein-Uhlenbeck (OU) semigroup from Gaussian analysis is a set of operators that we show to smoothly interpolate between a random variable and its expectation. The OU semigroup provides the appropriate set of smoothing operators which enable us to control variance and avoid posterior collapse.

We further show that by smoothing the intermediate parameterizing functions $\mu(z), \sigma(z)$ in the proposed manner, the KL divergence of the top layer sees a sudden sharp drop toward zero as the amount of smoothing is decreased. This behaviour is retained when we evaluate the KL divergence on the original *unsmoothed* variational autoencoder model. This behaviour is reminiscent of phase transitions from statistical mechanics and we adopt the same terminology to describe the phenomenon. Our experiments suggest that the phenomenon is general across datasets and commonly used architectures. Furthermore, the *critical value* of the smoothing parameter $\rho$ at which the transition occurs is fixed for a given model configuration and varies with stochastic depth and width.

We make the following contributions. First, we establish a connection between higher harmonics, variance, posterior collapse and phase transitions in hierarchical VAEs. Second, we show that by using the Ornstein-Uhlenbeck semigroup of operators on the generative stochastic functions in VAEs we reduce higher frequencies and consequently variance to mitigate posterior collpase. We corroborate our findings experimentally and further obtain in CIFAR-10 likelihoods competitive with more complex architectural solutions alongside a reduction in model size. We refer to the proposed family of models as *Hermite variational autoencoders* (HVAE).

## 2 HERMITE VARIATIONAL AUTOENCODERS

### 2.1 ANALYSIS ON GAUSSIAN SPACES

The analysis of Gaussian spaces studies functions of Gaussian random variables. These are real-valued functions defined on $\mathbb{R}^n$ endowed with the Gaussian measure. Many functions employed in machine learning are instances of such functions: decoders for variational autoencoders, as is the case in this work, and generators for generative adversarial networks being two examples.

By way of summary, the main facts we use from this field are that a function on a Gaussian space can be expanded in an orthonormal basis, where the basis functions are the Hermite polynomials. This orthonormal expansion is akin to a Fourier transform in this space. The second fact is that the coefficients of such an expansion can be modified in a way to reduce the variance of the expanded function by applying an operator from the Ornstein-Uhlenbeck semigroup of operators. Next, we give a brief introduction. For further details on Gaussian analysis we refer to [9].

**Gaussian Spaces:** Let $L^2(\mathbb{R}^n, \gamma)$ be the space of square integrable functions, $f : \mathbb{R}^n \to \mathbb{R}$, with the Gaussian measure $\gamma(z) = \prod_i \mathcal{N}(z_i|0, 1)$. Given functions $f, g$ in this space, the inner product is given by $\langle f, g \rangle = \mathbb{E}_{\gamma(z)}[f(z)g(z)]$.

**Basis functions for $L^2(\mathbb{R}, \gamma)$:** Taking the space of *univariate* functions $L^2(\mathbb{R}, \gamma)$ , it is known that the polynomial functions $\phi_i(z) = z^i$ are a basis for this space. By a process of orthonormalization we obtain the *normalized* Hermite polynomial basis for this space. The first few Hermite polynomials are the following: $h_0(z) = 1, \quad h_1(z) = z, \quad h_2 = \frac{z^2-1}{\sqrt{2}}, \ldots$.

**Basis functions for $L^2(\mathbb{R}^n, \gamma)$:** Letting $\alpha \in \mathbb{N}^n$ be a multi-index, the basis functions for $L^2(\mathbb{R}^n, \gamma)$ are obtained by multiplying the univariate basis functions across dimension, $h_\alpha(z) = \prod_i h_{\alpha_i}(z_i)$.

**Hermite expansion:** A function in $L^2(\mathbb{R}^n, \gamma)$ can be expressed as $f = \sum_{\alpha \in \mathbb{N}^n} \hat{f}(\alpha) h_\alpha$, where $\hat{f}(\alpha)$ are the Hermite coefficients of $f$ and are computed as $\hat{f}(\alpha) = \langle f, h_\alpha \rangle = \mathbb{E}_{\gamma(z)}[f(z) h_\alpha(z)]$. Plancherel's theorem is the following relation between the norm of $f$ and $\hat{f}$ which follows from orthnormality of the basis functions.

$$\langle f, f \rangle = \sum_\alpha \hat{f}(\alpha)^2, \tag{2}$$

**Ornstein-Uhlenbeck (OU) Semigroup:** Given a parameter $\rho \in [0, 1]$ and a Gaussian variable $z$, we construct a correlated variable $z'$ as $z' = \rho z + \sqrt{1 - \rho^2} z_\omega$, where $z_\omega \sim \mathcal{N}(0, 1)$ is a random standard Gaussian sample. The OU semigroup is a set of operators, denoted $U_\rho$ and parameterized by $\rho \in [0, 1]$. The action of $U_\rho$ on $f$ at $z$ is to average the function values on correlated $z'$s around $z$,

$$U_\rho f(z) = \mathbb{E}_{z'|z}[f(z')] = \mathbb{E}_{z_\omega}[f(\rho z + \sqrt{1 - \rho^2} z_\omega)] \tag{3}$$

The action of the $U_\rho$ operators on the Hermite expansion of function $f(z)$ is to decay Hermite coefficients according to their degree, $U_\rho f(z) = \sum_{\alpha \in \mathbb{N}^n} \rho^{|\alpha|} \hat{f}(\alpha) h_\alpha$. where $|\alpha| = \sum_i \alpha_i$.

If $z$ is reparameterized as $z = \sigma \epsilon_1 + \mu$, the correlated OU sample is given by $z' = \sigma(\rho \epsilon_1 + \sqrt{1 - \rho^2} \epsilon_2) + \mu$, where $\epsilon_1, \epsilon_2$ are standard Gaussian variables. This can also be expressed in terms of $z$ as

$$z' = \rho z + (1 - \rho)\mu + \sigma \sqrt{1 - \rho^2} \epsilon_2, \tag{4}$$

## 2.2 Hermite expansions for VAEs

Our proposed method is a new training procedure for the vanilla hierarchical variational autoencoder that builds upon Hermite expansions of Gaussian functions and properties of the OU semigroup.

In the context of hierarchical variational autoencoders, the Gaussian function $f$ is the generative model $\mu_i(z_{i+1})$ and $\sigma_i(z_{i+1})$ that receives as inputs the latent variable $z_{i+1}$ to return the Gaussian latent variable of the next layer, $z_i \sim \mathcal{N}(\mu_i(z_{i+1}), \sigma_i(z_{i+1}))$.

We make use of the following properties of the OU semigroup to construct Gaussian functions of lower variance. The first property we employ is that the OU semigroup of operators interpolates between a random variable ($\rho = 1$) and its expectation ($\rho = 0$), where the parameter $\rho$ controls the extent of the interpolation.

**Proposition 1** *The operators $U_\rho$ retain the expected value of the operated function, $\mathbb{E}[f] = \mathbb{E}[U_\rho f]$.*

**Proposition 2** *The operators $U_\rho$ interpolate between a random variable and its expectation. In particular, as $\rho \to 1$, $U_\rho f = f$. and as $\rho \to 0$, $U_\rho f = \mathbb{E}[f]$*

The second property we exploit is that the new random variable $U_\rho f(z)$ has lower variance compared with original variable $f(z)$ and is in general a smoother function than $f(z)$. The smoothing properties of the operator $U_\rho$ can be understood by examining the Hermite expansion of $U_\rho f$. First we note that we can express the expectation and variance of a function $f$ in terms of its Hermite coefficients, specifically $\mathbb{E}[f] = \hat{f}(0)$ and $\mathrm{Var}(f) = \mathbb{E}[(f - \mathbb{E}[f])^2] = \mathbb{E}[(f - \hat{f}(0))^2] = \sum_{\alpha:|\alpha|>0} \hat{f}(\alpha)^2$, which follows from Plancherel's theorem (equation 2).

Replacing $f$ with $U_\rho f$ and using the Hermite expansion of $U_\rho f$ from equation 3, the mean remains the same, $\mathbb{E}[U_\rho f] = \rho^0 \hat{f}(0) = \hat{f}(0)$, and variance reduces like

$$\mathrm{Var}[U_\rho f] = \mathbb{E}[(U_\rho f - \mathbb{E}[f])^2] = \mathbb{E}[(f - \hat{f}(0))^2] = \sum_{\alpha:|\alpha|>0} \rho^{2|\alpha|} \hat{f}(\alpha)^2. \tag{5}$$

The last equation indicates that the contribution to the variance by $\hat{f}(\alpha)$ decays by an amount $\rho^{2|\alpha|}$ when $\rho \in (0, 1)$. This, in turn, leads to a decrease in variance.

**Algorithm.** In essence, Hermite variational autoencoders are similar to variational autoencoders, save for applying the OU semigroup to the latent distributions $p(z_i|z_{i+1})$ that comprise the generator to compute gradients during *training* only. Specifically, we apply these operators to the functions parameterizing the mean and variance of the latent Gaussian distributions. For each distribution $p(z_i|z_{i+1})$ we substitute $\mathcal{N}(z_i|\mu_i(z_{i+1}), \sigma_i(z_{i+1}))$ with $\mathcal{N}(z_i|U_\rho\mu_i(z_{i+1}), U_\rho\sigma_i(z_{i+1}))$. The new functions result in latent distributions with parameters that have lower variance but the same expected value relative to the conditional input latent distribution.

In an alternative parameterization we apply the OU semigroup to the ratio of the mean and variance functions: $U_\rho\frac{\mu_i}{\sigma_i}(z_{i+1})$ (see next section for a justification of this). The OU semigroup operators can also be applied on approximate posterior functions, but we observe little benefit. In practice, we compute $U_\rho\mu_i(z_{i+1})$ and $U_\rho\sigma_i(z_{i+1})$ by Monte Carlo averaging. As for a function $f$, $U_\rho f = \mathbb{E}_{z'|z}[f(z')]$, where $z'$ are the correlated samples, we estimate the expectation by Monte Carlo averaging over $z'$. Experiments show that 5 to 10 samples suffice.

It is important to emphasize that the substitution of the lower variance functions for parameterizing the distributions is *only done when computing gradients during training*. All evaluations, training or test, are still done on the original hierarchical variational autoencoder model. Thus, the new training procedure has an additional computational cost only for the intermediate distributions in the generator, proportional to the number of correlated samples during training.

**Complexity.** In Hermite VAE the OU sampling operation is only applied in the intermediate stochastic layers in the generator network. In particular, it is not applied in the inference network or in the last layer of the decoder. The fact that OU sampling is not applied in the final stochastic layer computing $p(x|z_1)$ is especially important for deep VAEs for images since feature maps are upsampled to match image dimensions in this layer. Thus, for 5 OU samples, the added computational and activation memory complexity is significantly less than 5 times the total cost of the base VAE model, and is 5 times the cost in the higher decoder layers only in the base model. An empirical comparison of maximum memory usage of various models can be found in table 6.

## 3 KL DIVERGENCE ANALYSIS

In this section we justify our approach as affecting a bias-variance trade-off in the KL divergence terms of the hierarchical VAE objective. The bias-variance trade-off arises from the fact that the KL divergence term can be written so that it contains the variance of the functions $\mu(z), \sigma(z)$ parameterizing the intermediate distributions. This variance is related to spectral complexity and reducing spectral complexity leads to a reduction in variance.

Consider the following expression from the ELBO as part of the KL divergence term.

$$\mathbb{E}_{q(z_1, z_2|x)}[\log p(z_1|z_2)] = \mathbb{E}_{q(z_1|x)}\mathbb{E}_{q(z_2|z_1)}[\log p(z_1|z_2)] \tag{6}$$

where $p(z_1|z_2) = \mathcal{N}(z_1|\mu_p(z_2), \sigma_p)$ and $q(z_1|x) = \mathcal{N}(z_1|\mu_q(x), \sigma_q(x))$. For the purpose of analysis we assume that the standard deviation for $p$ is fixed and independent of $z_2$, $\sigma(z_2) = \sigma_p$. Furthermore for ease of analysis and without loss of generality we think of $z_1$ as a scalar. The general result for multivariate $z_1$ follows from summing for all dimensions.

For Gaussian $p$ we write equation 6 as $\mathbb{E}_{q(z_1, z_2|x)}[-\log\sqrt{2\pi\sigma_p^2} - \frac{1}{2\sigma_p^2}(z_1 - \mu_p(z_2))^2]$. From the inner term $\mathbb{E}_{q(z_1, z_2|x)}[(z_1 - \mu_p(z_2))^2]$ we focus on the quadratic $\mu_p(z_2)^2$ which is expanded as

$$\frac{1}{2\sigma_p^2}\mathbb{E}_{q(z_1, z_2|x)}[\mu_p(z_2)^2] = \frac{1}{2\sigma_p^2}\mathbb{E}_{q(z_1|x)}[\mathbb{E}[\mu_p(z_2)]^2 + \text{Var}(\mu_p)] \tag{7}$$

By Plancherel's theorem (equation 2) we have

$$\frac{1}{2\sigma_p^2}\mathbb{E}_{q(z_1, z_2|x)}[\mu_p(z_2)^2] = \frac{1}{2\sigma_p^2}\mathbb{E}_{q(z_1|x)}\left[\hat{\mu_p}(0)^2 + \sum_{\alpha:|\alpha|>0}\hat{\mu_p}(\alpha)^2\right]. \tag{8}$$

This shows that for $\sigma_p$ independent of $z_2$ the KL divergence term in the ELBO contains the variance of the parameterizing function $\mu_p(z_2)$.

In our proposal we replace $\mu_p(z_2)$ by $U_\rho[\mu_p(z_2)]$ and the right side of equation 8 becomes

$$\frac{1}{2\sigma_p^2}\mathbb{E}_{q(z_1|x)}[\mathbb{E}[U_\rho\mu_p]^2 + \text{Var}(U_\rho\mu_p)] = \frac{1}{2\sigma_p^2}\mathbb{E}_{q(z_1|x)}\left[\hat{\mu}_p(0)^2 + \sum_{\alpha:|\alpha|>0}\rho^{2|\alpha|}\hat{\mu}_p(\alpha)^2\right] \quad (9)$$

since $\mathbb{E}[U_\rho f] = \mathbb{E}[f]$. The new variance is of order $O(\rho^2)$. Comparing this objective with the original VAE objective, for the second term we get a bias proportional to the difference of the variance

$$\text{bias} = \frac{1}{2\sigma_p^2}(\text{Var}(\mu_p) - \text{Var}(U_\rho\mu_p)).$$

Comparing equations 8 and 9 we see the bias to be $O(1 - \rho^2)$.

The analysis above assumed $\sigma_p$ independent of $z_2$. For $\sigma_p$ dependent on $z_2$ we can make a similar argument for the variance of $(z_1 - \mu(z_2))/\sigma(z_2)$ or for the ratio $\mu(z_2)/\sigma(z_2)$ by expanding the square and considering the terms separately.

## 4 RELATED WORK

To increase the stochastic depth of VAEs [10, 19], [21] proposes the Ladder VAEs. With an architecture that shares a top-down dependency between the encoder and the decoder, Ladder VAEs allow for interactions between the bottom-up and top-down signals and enable training with several layers deep VAEs. Extending Ladder VAEs, [13] proposed the bidirectional-inference VAEs, adding skip connections in the generative model and a bidirectional stochastic path in the inference model.

[1] observed that the latent distribution collapses to the prior in deep stochastic hierarchies – a phenomenon now called *posterior collapse*. Posterior collapse appears in different contexts including images or text, and is strongly associated with the presence of powerful decoders, be it LSTMs [1] for text or strong autoregressive models for images [16], where although the model may produce good reconstructions, it does not learn a meaningful generative distribution. A prevalent hypothesis behind posterior collapse is that when the decoder is strong enough to generate very low cross entropy losses, the optimization may find it easier to simply set the KL divergence term to 0 to minimize the ELBO [1]. Making an association with probabilistic PCA, [12] hypothesize that posterior collapse is caused by local optima in the optimization landscape due to high variance, even without powerful decoders. High variance was identified as a potential culprit also by [21] for posterior collapse.

Given the breadth of the problem, many tried to address posterior collapse. [1, 8, 21, 14] anneal the KL divergence between the approximate posterior to the prior from 0 to 1. Unfortunately, this solution does not optimize the original ELBO formulation and is shown [27, 3] to cause instabilities, especially with large datasets and complex decoders. [11] introduce the concept of *free bits* ignoring the gradient if not significant enough. [17] proposed $\delta$-VAEs, which constrain the latent distribution to have a minimum distance to the prior. [7] monitor the mutual information between the latent and the observed variable to aggressively optimize the inference model before every model update. [2] suggest that using a tighter multi-sample ELBO can help alleviate collapse to some extent.

While many [1, 21] suggested a connection between posterior collapse and variance reduction, no real solutions using variance reduction has been proposed. One reason may be the low variance the reparameterization trick [10, 19] already offers. Empirically, while the reparametrization is successful with producing low variance forward and backward propagations in shallow models, it has not been enough for deeper and wider ones. Another reason suggested by the approach of this paper is that variance appears as a side effect of spectral complexity.

To reduce the variance in reparameterization gradients, [20] suggest removing a mean zero score function term from the total derivative while [15] build a control variate using a linear approximation for variance reduction. [2] propose importance weighted gradients and [24] extend [15] to multiple samples to obtain an estimator with improved signal-to-noise ratio. Other approaches to increase the power of VAE models include normalizing flows [18], better posterior or prior distributions [22], adding autoregressive components [6] or a combination of both [11].

Here, we theoretically argue and empirically validate that damping higher frequency components, thus lowering variance, allows for training deeper latent hierarchies while addressing posterior collapse.

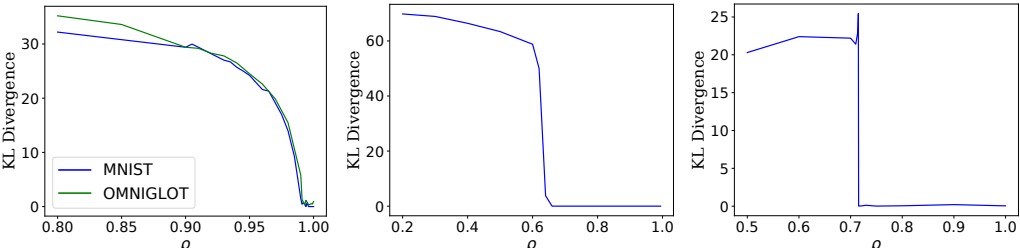

Figure 1: VAE KL divergence undergoes as a phase transition as the model as we decrease the amount of smoothing to recover the original model at $\rho = 1$. The figure shows top layer KL divergence *vs.* $\rho$, where each point is a different model after 100000 steps of training with a fixed $\rho$. The plots show the KL divergence values for the original unsmoothed model after training on the smoothed version. The left figure shows this for same 4 stochastic (40 units per layer) layer MLP model on MNIST and OMNIGLOT. It can be seen that the critical value where the KL divergence drops toward 0 is the same for both datasets on this model. The next two figures show phase transition for a 3 stochastic layer (200 units per layer) MLP model on OMNIGLOT (middle) and a convolutional model (8x7x7 latent dimension) on MNIST (right) showing that the phase transition becomes especially pronounced with the width of the latent layer.

We rely on tools from the field of analysis on Gaussian spaces, amenable to the analysis of stochastic processes [9].

## 5 EXPERIMENTS

In the following experiments we first test our method's ability to prevent posterior collapse and compare with other methods designed for the same end. We then present our observation that the top layer KL divergence undergoes a phase transition as we decrease the amount of smoothing. Finally, we compare performance on commonly used benchmarks against other methods from the literature.

We validate Hermite variational autoencoders on binary MNIST, OMNIGLOT and CIFAR-10 with various ResNet and MLP architectures. Validation ELBOs are evaluated using importance-weighted samples [2] with $\mathcal{L}_{100}$ and $\mathcal{L}_{5000}$ denoting evaluation with 100 and 5000 samples.

### 5.1 INVESTIGATING POSTERIOR COLLAPSE

We test Hermite VAEs for their ability to prevent posterior collapse on basic MLP network architectures. We compare on static and dynamically binarized MNIST against various standard methods for mitigating posterior collapse. For dynamic MNIST we choose two models: the first is a 4 stochastic layer model with 64,32,16,8 latent variables. The deterministic layers have two layers of 512,256,128,64 units respectively, going from bottom to top. The second model has 4 stochastic layers VAE with 40 units per stochastic layer and 2 layers of 200 units per stochastic layer . For static MNIST we only use the second model described above. All models have a simple stacked architecture with no skip connections.

First we compare with a standard VAE on static MNIST. We trained the VAE with the standard training method and our method with $\rho \in \{0.9, 0.8\}$. We show the validation curves and the KL divergence of the top stochastic layer $KL(q(z_4|z_3)||\mathcal{N}(0,1))$ in figure 2. The standard training collapses the posterior immediately while our method avoids posterior collapse and yields better validation ELBO.

Table 1: Test ELBO on statically binarized MNIST with MLP

| Model | ELBO |
|---|---|
| VAE (L=2) | -86.05 |
| VAE (L=1)+NF [18] | -85.10 |
| IWAE (L=2) [2] | -85.32 |
| VampPrior (L=2) [23] | -83.19 |
| HVAE (L=4), $\mathcal{L}_{5000}$ | -83.42 |

Table 2: Test ELBO on dynamically binarized MNIST with MLP

| Model | ELBO |
|---|---|
| Ladder VAE (L=5) | -81.7 |
| VampPrior (L=2) [23] | -81.24 |
| HVAE (L=4), $\mathcal{L}_{5000}$ | -81.2 |
| HVAE (L=5), $\mathcal{L}_{5000}$ | -81.1 |

Table 3: Posterior collapse on dynamic MNIST. The table shows top layer KL divergence and active units (top-to-bottom) for various method on the 4 stochastic layer models. The 64-32-16-8 latent dimension models have two layers of 512, 256, 128, 64 hidden units in each stochastic layer respectively. The 40-40-40-40 latent dimension models have two layers of 200 units per stochastic layer. '+KL' indicates KL annealing. All models were trained for 1M steps with the same hyperparameters.

| Model | V. ELBO ($\mathcal{L}_{100}$) | Reconstruction | KLD | Top KLD | Active Units |
|---|---|---|---|---|---|
| IWAE (64-32-16-8) | -84.46 | -65.3 | 23.98 | 4.88 | 3,15,30,64 |
| IWAE (40-40-40-40) | -84.63 | -65.5 | 23.98 | 1.18 | 0,4,37,40 |
| VAE+KL (64-32-16-8) | -84.6 | -60.3 | 28.8 | 6.2 | 6,11,25,49 |
| VAE+KL (40-40-40-40) | -84.7 | -60.8 | 28.07 | 1.13 | 1,6,15,40 |
| VAE+Freebits (64-32-16-8) | -85.5 | -64.2 | 25.1 | 3.8 | 2,4,9,21 |
| VAE+Freebits (40-40-40-40) | -86.0 | -65.8 | 23.6 | 2.46 | 1,2,8,18 |
| HVAE ($\rho = 0.95$) (64-32-16-8) | -81.6 | -59.78 | 26.1 | 8.99 | 8,16,32,54 |
| HVAE ($\rho = 0.9$) (64-32-16-8) | -81.7 | -60.0 | 25.6 | 9.56 | 8,16,32,43 |
| HVAE ($\rho = 0.95$) (40-40-40-40) | -84.4 | -65.7 | 23.7 | 9.34 | 40,40,40,40 |

Table 4: Posterior collapse on static MNIST. The table shows top layer KL divergence and active units (top-to-bottom) for various method on the same 4 stochastic layer model with 40 units each, '+KL' indicates KL annealing. All models were trained for 1M steps with the same hyperparameters.

| Model | V. ELBO ($\mathcal{L}_{100}$) | Top KLD | Active Units |
|---|---|---|---|
| VAE | -86.4 | 0.82 | 1,3,9,17 |
| IWAE (5 samples) | -84.9 | 1.005 | 1,3,9,24 |
| VAE+KL | -84.5 | 1.2 | 2,3,11,37 |
| VAE+Freebits | -87 | 1.3 | 1,1,8,17 |
| HVAE ($\rho = 0.85$) | -85.2 | 20.6 | 40,40,40,24 |
| HVAE ($\rho = 0.9$) | -85.5 | 19.5 | 40,40,40,40 |
| HVAE+KL ($\rho = 0.95$) | -84.2 | 23.5 | 38,34,17,40 |

Next we compare against other methods designed to mitigate posterior collapse including KL annealing, free bits [11] and importance weighted objectives [2]. For KL annealing the annealing coefficient is set to 0 for the first 10,000 steps and is linearly annealed to 1 over the next 500,000 steps. For free bits, we apply the same free bits value to each stochastic layer. The free bits values are chosen from $\{0.5, 1.0, 2.0, 3.0\}$. We find that training slows down considerably when using free bits. Values of free bits of 4.0 or more caused training to become unstable.

For this comparison we use the 40-40-40-40 4-layer architecture used for all methods for static MNIST and both the 40-40-40-40 and 64-32-16-8 architectures for dynamic MNIST. We show the results in tables 4 and 3, where we include the total KL divergence, top layer KL divergence as well as the number of active units in each of the 4 layers. Compared to the baseline methods, our method is able to maintain significant activity across the layers and a considerably higher KL divergence in the top layer. For dynamic MNIST the 64,32,16,8 model has successively smaller networks higher in the hierarchy and the other methods show better activity in the latent units than the 40-40-40-40 model. This suggests that training dynamics and the architecture affect the extent of posterior collapse with standard methods and have less of an effect with our proposal.

Here we also find that employing other posterior collapse mitigation techniques can sometimes help with HVAE as well. As shown in table 4, KL annealing with the 4 layer HVAE allows it to reach higher validation ELBO. However, we did not find KL annealing to be advantageous when used with HVAE on more complex architectures and in particular we do not see improvement in CIFAR-10 performance.

### 5.1.1 PHASE TRANSITIONS

We have given some evidence to show that attenuating the higher frequency components of parameterizing functions by smoothing, thus also reducing variance, is a justifiable mitigation against posterior collapse.

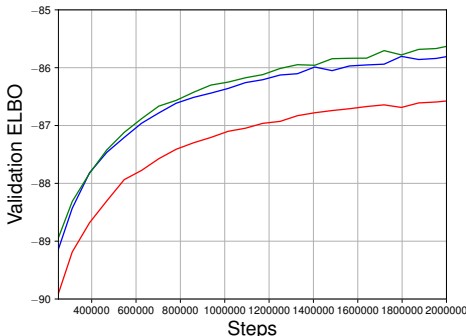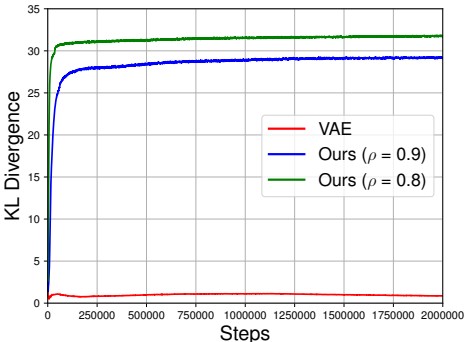

Figure 2: Training 4 layer VAE models on static MNIST with validation ELBO on the left and $KL(q(z_4|z_3)||\mathcal{N}(0,1))$ on the right. The VAE shows posterior collapse, while HVAE avoids it alongside improved validation ELBO.

Here we show that as the amount of smoothing is reduced (as $\rho$ approaches 1) to recover the original VAE gradients, the top level KL divergence shows a sudden decline at a critical value of the smoothing parameter. The sharpness of this decline depends on the model configuration (stochastic layers, latent dimension). On the other hand, our experiments suggest that the sharpness of the decline is independent of dataset.

An example of this phenomenon can be seen in figure 1 and the appendix. Here we show the top layer KL divergence after training for 100k steps with varying $\rho$ for 3 different architectures. We see that the phase transition becomes more pronounced with greater stochastic dimension. Figure 3 further shows how HVAE can prevent collapse across a spectrum of varying stochastic depth and width while maintaining good validation performance.

Table 5: Comparing bits per dimension, parameter efficiency and model depths between hierarchical methods and other well performing methods on CIFAR-10. '+' indicates stochastic skip connections.

| Model | BPD | Layers | Parameters |
|---|---|---|---|
| LVAE [13] | 3.60 | 15 | 72.36M |
| LVAE+ [13] | 3.41 | 15 | 73.35M |
| LVAE+ [13] | 3.45 | 29 | 119.71M |
| BIVA [13] | 3.12 | 15 | 102.95M |
| Discrete VAE++ [26] | 3.38 | – | – |
| NICE [4] | 4.48 | – | – |
| RealNVP [5] | 3.49 | – | – |
| NVAE [25] | 2.91 | – | – |
| VAE+IAF [11] | 3.11 | – | – |
| HVAE, $\mathcal{L}_1$ | 3.5 | 3 | 9.95M |
| HVAE, $\mathcal{L}_1$ | 3.46 | 4 | 12.4M |
| HVAE, $\mathcal{L}_1$ | 3.43 | 6 | 16.8M |
| HVAE+, $\mathcal{L}_1$ | 3.42 | 3 | 9.95M |
| HVAE+, $\mathcal{L}_{100}$ | 3.39 | 3 | 9.95M |

## 5.2 BENCHMARK COMPARISONS

### 5.2.1 MNIST & OMNIGLOT

**MLPs.** For static MNIST we report test ELBO for MLPs with 4 stochastic layers with 40 units each. Before each stochastic layer we have two deterministic layers of 200 tanh units.

We show the results in table 1 comparing with others that improve VAEs: IWAE [2], VAE with normalizing flow [18] and VampPrior [23].

The dynamic MNIST results are in table 2. The 5-layer HVAE model in this instance has latent dimensions 64, 32, 16, 8, 4 with two layers of 512, 256, 128, 64, 32 units in the respective stochastic layers.

**Residual ConvNets.** Next, we experiment with a more complex ResNet architecture on MNIST and OMNIGLOT with up to 4 stochastic layers and up to 5 ResNet blocks between stochastic layers ($14 \times 14$ features maps). The networks have the same hierarchical structure as the MLP VAEs with convolutional latent layers. We do not employ stochastic skip connections between blocks.

We improve the MLP scores reaching -96.08 validation ELBO on OMNIGLOT, compared to -97.65 and -97.56 for VAE(L=2) and VampPrior (L=2). The detailed results are in the appendix.

In both experiments we obtain competitive scores despite the simple architecture.

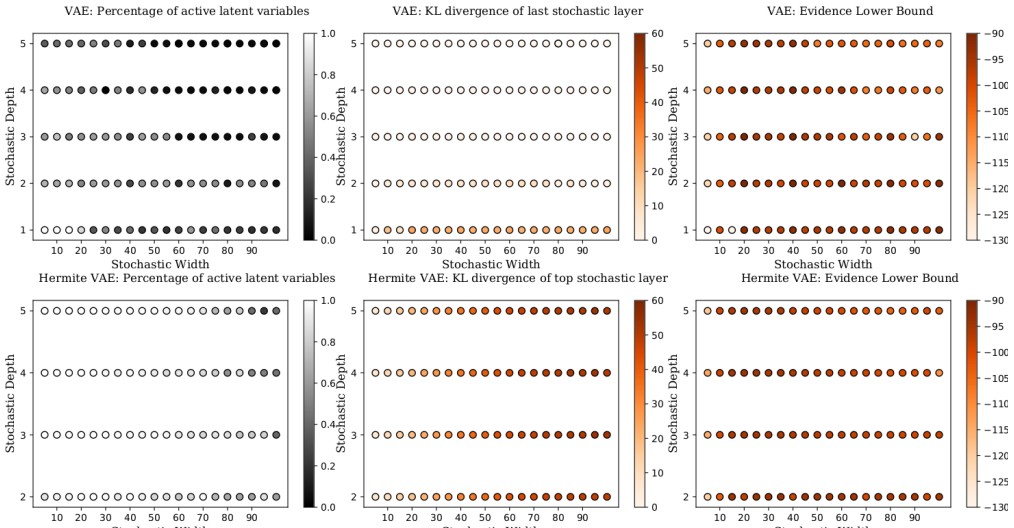

Figure 3: VAEs (above) exhibit suddenly many inactive units when increasing the depth to 2 layers and the width beyond 20 neurons, while collapsing completely after 3 layers and 60 neurons. The proposed Hermite VAEs (below) retain for nearly all cases full latent activity, while maintaining similar ELBOs (not shown). Trained on MNIST for 100K steps for practical reasons (normal convergence if left training), with 2 deterministic, 200-neuron $\tanh$ layers before every stochastic layer.

### 5.2.2 CIFAR-10

We experiment with ResNets of up to 6 stochastic layers, interwined with deterministic layers comprising 6 ResNet blocks. We used 100 feature maps for all deterministic layers. The stochastic layers have 8 feature maps of width $16 \times 16$. We also experiment with skip connections between stochastic layers. We report results in table 5. We obtain an ELBO of 3.5 bpd without skip connections. After adding skip connections, we improve to 3.42 bpd on the 3-layer architecture. This corresponds to an ELBO of 3.39 when evaluated with 100 importance samples. This result is on par with the 15-layer Ladder VAE (LVAE) [13], and comparable to the 15-layer LVAE+ [13] architecture that adds skip connections to LVAE.

Note also that the improvement comes at a signification reduction of parameters. Compared to Ladder VAE the 3 layer Hermite VAE uses about 7 times fewer parameters.

Table 6: Comparing maximum memory usage for various models. We report the memory usage 4 and 6-layer models on CIFAR-10, using 5 deterministic layers per stochastic layer with 100 units per layer and 5 OU or IWAE samples. The memory usage is the maximum amount of used memory with a batch size of 64. For LVAE and BIVA we use code from the BIVA [13] PyTorch repository.

| Model | Layers | Memory | Parameters (M) |
|-------|--------|--------|----------------|
| VAE   | 4      | 2.12G  | 12.4           |
| VAE   | 6      | 2.45G  | 16.8           |
| IWAE  | 4      | 8.59G  | 12.4           |
| IWAE  | 6      | 10.5G  | 16.8           |
| LVAE  | 15     | 6.4G   | 59.5           |
| BIVA  | 15     | 10.3G  | 103            |
| HVAE  | 4      | 3.74G  | 12.4           |
| HVAE  | 6      | 4.6G   | 16.8           |

## 6 CONCLUSION

Training variational autoencoders with large hierarchies of stochastic layers has proven to be difficult. An often shared observation when moving to more complex variational autoencoders is posterior collapse where a subset of the latent variables falls back to the prior distribution.

For a solution we turn to the field of analysis of Gaussian functions and analyze intermediate VAE functions as Hermite expansions. We argue that the Ornstein-Uhlenbeck semigroup can be used to reduce variance and empirically show that its parameterizing variable can be used to precisely control phase transitions. We validate the analysis and solution on three datasets, MNIST, OMNIGLOT, CIFAR-10, where we are able to avoid posterior collapse and obtain performance that is competitive with more complex architectural solutions.

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
