# OpenReview forum: "Variance Reduction in Hierarchical Variational Autoencoders"
_ICLR.cc/2021/Conference — Reject_

### Official Review · AnonReviewer1 · 2020-10-24
**Idea is sensible, but analysis is incomplete**

**Rating:** 4
**Confidence:** 4

**Review:**

Post-rebuttal update
---------

Thank you for your response.  Now I understand that the algorithm works by smoothing the Gaussian parameters $\mu_i,\sigma_i$ w.r.t. the centered Gaussian rv (as described in my last reply, second part of bullet point (1)), so my original concern regarding the bias _in the Gaussian parameters_ does not hold.  However, I still cannot recommend acceptance at this point, because of a newly discovered issue in the theoretical analysis:

The analysis in Section 3 does not take into account the impact of smoothing on the ``downstream'' nonlinear layers.  The text only considers two layers of stochastic latents and the KL part of ELBO, but in the deeper case, the smoothing of $\mu_i(z_{i+1})$ will additionally have influence on the layers below $i$, through the nonlinear functions $\mu_{i'},\sigma_{i'}$ for $i'<i$.  More concretely, consider the following scenario: $\mu_i(z_{i+1})\equiv z_{i+1}, \sigma_i(z_{i+1})\equiv \epsilon$ which is very small. Further assume that $z_{i+1}$ is high-dimensional and approximately follows $\mathcal{N}(0,I)$, so $\\|z_i\\|_2 = \\|\mu_i(z_{i+1})+\sigma_i\varepsilon_i\\|_2 \approx \\| z_{i+1} \\|_2 > 100$ with probability $1-\epsilon_1$, where $\epsilon_1$ is also very small. In this case, it is possible to achieve a low KL in the original ELBO, by using a $\mu_{i-1}$ which only has sensible values in the region $B := \\{z_i: \\|z_i\\|>100\\}$; in the complement set $B^c$, $\mu_{i-1}$ can be "arbitrarily" bad so long as its impact on the ELBO does not outweigh $\epsilon_1$, the probability its input falls there. However, in the smoothed estimator with $\rho=0$, the input to $\mu_{i-1}$ only have norm $O_p(\sigma_i(z_{i+1}))=O_p(\epsilon)$, so the value of $\mu_{i-1}$ on $B^c$ will have a far higher impact, easily exceeding the original by $O(1/\epsilon_1)$.  To summarize, *it is possible to construct models where the ELBO has a reasonable value, but the smoothed objective behaves catastrophically*.  Moreover, even in the shallow case, $z_i$ will be fed into a final decoder block to generate the reconstruction image, so a similar issue exists, although it will be in the reconstruction likelihood part of the ELBO as opposed to the KL part.

A less important issue is that parts of the analysis are written in a confusing way.  Apart from the abuse of notation $U_\rho$ which leads to my original confusion, in Section 3 the $\hat{\mu}_p$'s should have a suffix of $z_1$, to signify the fact that they are coefficients of a function that depends on $z_1$ (see the last response from he authors).  Also it is unclear to me why there is no mention of $\mu_p^4$, in the analysis of the variance of an estimator for $\mu_p^2$.  But given the aforementioned issue, I don't think it is necessary to look further into this case.


Summary
-------

This work proposes to smooth the mean and variance parameters in the decoder of hierarchical VAEs with the O-U process. It is shown that the smoothing procedure reduces variance of the ELBO, alleviates posterior collapse, and improves on model likelihood on CIFAR-10 under a fixed number-of-parameter budget.

The idea to investigate the impact of ELBO variance in hierarchical VAE performance is sensible, and the experiments seem to show improvements. However, I have concerns regarding the theoretical claims, and the empirical results also seem to need clarification.

Major Concerns
--------------

- The claim that the smoothing doesn't change the expectation (of functions acting on the latents) doesn't seem correct. Prop.1 and 2 only holds when the expectation is taken wrt the standard normal distribution, while all but the top-level latents (i.e., $z_i$ for $i<L$) come from a mixture of Gaussian. Intuitively it also seems incorrect: what if $\rho=0$?

- The variance analysis works by assuming $\sigma_q$, the decoder variance, is constant. This ignores the problem of unbounded likelihood [1], where posterior variance goes to zero, thus driving the ELBO and its variance to infinity. It would be helpful to include a plot of the decoder variances in the most realistic model, to see if this issue is relevant in modern hierarchical VAEs (and thus whether the analysis here provides a complete picture).

- The conclusion of the analysis does not seem helpful: the bias is $O(1-\rho^2)$ and the variance is $O(\rho^2)$, so it is unclear from the bound whether there will be a $\rho$ that decreases the overall MSE.

Minor
--------------

- It is worth mentioning that there are several types of posterior collapse and not all of them are undesirable [2]: sometimes it is superfluous units rightfully pruned [3, 4]. This also implies that the number of active units is not a good measure of model quality; it is helpful to include reconstruction error in Section 5.1.

- The observed phase transition of KLD connects to the fact that ELBO-trained VAE acts like a thresholding operator; see [2].

- Why didn't Table 3 mention NVAE [5] and IAF-VAE [6], both of which have better BPD values? Seeing where those models are on the #parameters-BPD curve helps to put the results here in perspective.

References
----------

[1]: Mattei and Frellsen, Leveraging the Exact Likelihood of Deep Latent Variable Models, in NeurIPS 18.

[2]: Dai et al, The Usual Suspects? Reassessing Blame for VAE Posterior Collapse, in ICML 20.

[3]: Lucas et al, Don't Blame the ELBO! A Linear VAE Perspective on Posterior Collapse, in NeurIPS 19.

[4]: Dai and Wipf, Diagnosing and Enhancing VAE Models, in ICLR 19.

[5]: Vahdat and Kautz, NVAE: A Deep Hierarchical Variational Autoencoder, in NeurIPS 20.

[6]: Kingma et al, Improved variational inference with inverse autoregressive flow, in ICLR 16.

---

> ### Author Response · Authors · 2020-11-19
> **Response (part 1)**
>
> We appreciate the comments and attempt to address the concerns below.
>
> ##Major
> >The claim that the smoothing doesn't change the expectation (of functions acting on the latents) doesn't seem correct. Prop.1 and 2 only holds when the expectation is taken wrt the standard normal distribution, while all but the top-level latents (i.e.,... for ... ) come from a mixture of Gaussian. Intuitively it also seems incorrect: what if ... ?
>
>
> The reviewer is correct in that the marginal distribution, $p(z_i)$, of the incoming latent variables is a mixture of Gaussians. However, in the analysis we only work with conditional distributions, $p(z_i|z_{i+1})$.
> To conclude, the smoothing by $\rho$ does not change the expectation and the propositions hold.
> In that case for $\rho=0$, the output is the expected value of the function.
>
> We have made a clarification in the text on page 4 paragraph 1.
>
> >The variance analysis works by assuming ..., the decoder variance, is constant. This ignores the problem of unbounded likelihood [1], where posterior variance goes to zero, thus driving the ELBO and its variance to infinity. It would be helpful to include a plot of the decoder variances in the most realistic model, to see if this issue is relevant in modern hierarchical VAEs (and thus whether the analysis here provides a complete picture).
>
> There is a miscommunication here. To understand posterior collapse, our analysis focuses only on the KL divergence term of the ELBO. It makes no reference to the output model of the decoder of the generative model.
>
> To address the core of the question, we note that the problem of unbounded likelihood depends on the output model for the decoder.
> As mentioned in the paper cited by the reviewer, for instance, Gaussian output models face the problem of unbounded likelihoods while Bernoulli output models do not. Either case does not affect our analysis. To conclude, the bearing of the decoder on unbounded likelihoods is definitely important for any VAEs but we focus on posterior collapse in the KL term specifically.
>
> In VAE implementation it is common practice to constrain the variance when using Gaussian decoders. That constraining the variance prevents unbounded likelihoods is justified by proposition 2 of the cited paper (Mattei and Frellsen, 2018).
> Nevertheless to check whether realistic VAEs might suffer from this problem when variance is left unconstrained, we used a ResNet decoder VAE on CIFAR with Gaussian output with an unconstrained variance and track the minimum variance over dimensions during training.
> We add a plot of the minimum decoder output variance alongside training and validation ELBO with the Gaussian decoder on CIFAR on Appendix A.5. We observe that even when unconstrained the variance remains bounded away from 0 and that training and validation ELBOs are quite similar.
>
> >The conclusion of the analysis does not seem helpful: the bias is .. and the variance is .. , so it is unclear from the bound whether there will be a
> that decreases the overall MSE.
>
> The usefulness of the analysis is in justifying the smoothing operation of OU sampling, showing that it reduces variance at the cost of bias.
>
> The analysis does not focus on the output likelihood, only the KL term to analyze the effect of our method in addressing posterior collapse.
> The analysis does not derive, therefore, a single value of the parameter $\rho$ that provably gives the best bound.
> Rather our intent is to take the analysis in conjunction with the experimental results.
>
> Experiments indeed show a consistent behavior and phase transitioning as we change $\rho$.
> Specifically, experiments corroborate that there exists a single value of $\rho$ for which models consistently undergoes posterior collapse for different datasets given the same architecture. To derive an optimal $\rho$ that provably gives prevents collapse we would require analysis of the decoder. This is a good suggestion. As it is out of scope of the current paper, which focuses on posterior collapse, we leave the suggestion for future work.

---

> > ### Author Response · Authors · 2020-11-19
> > **Response (part 2)**
> >
> > ##Minor
> > > It is worth mentioning that there are several types of posterior collapse and not all of them are undesirable [2]: sometimes it is superfluous units rightfully pruned [3, 4]. This also implies that the number of active units is not a good measure of model quality; it is helpful to include reconstruction error in Section 5.1.
> >
> > This is certainly true.
> > This is why we had included validation ELBO alongside the active unit numbers (table 2 in update) showing that the model is learning something useful.
> > Based on your comment along with results on dynamic MNIST we now show reconstruction, total KL divergence and top KL divergence (table 1) for multiple methods and architectures.
> > We also have a new experiment in appendix A.4 (see also relevant response to reviewer 2) to test whether the low activity is due to pruning. There we vary the latent dimension from large to small but find the same low activity in the top layer. At the same time the validation ELBO drops significantly. Had the VAE model been having posterior collapse as a means of feature selection (at least in the top layers), we would expect better activity with the poorer validations ELBOs. And in reverse, if our Hermite VAE would harm the ability of the model to do feature selection we would expect no improvements on the validation ELBOs. For more details please refer to the answer to the first review: https://openreview.net/forum?id=uvEgLKYMBF9&noteId=QVwPp9Utaf6.
> >
> > To conclude, there are no signs in the experiments of neither using posterior collapse for feature selection pruning in the higher layers nor that the proposed Hermite VAE harms the ability of feature selection via posterior collapse pruning. This does not discount the possiblity of pruning in lower layers. For these lower layers closer to the output (pixels) experiments show that VAE does a better job of retaining information, and also HVAE lets neurons ``'die' when deemed necessary.
> >
> > > The observed phase transition of KLD connects to the fact that ELBO-trained VAE acts like a thresholding operator; see [2].
> >
> > Thank you for the suggestion. This does seem related. We will update the paper with the reference.
> >
> > > Why didn't Table 3 mention NVAE [5] and IAF-VAE [6], both of which have better BPD values? Seeing where those models are on the #parameters-BPD curve helps to put the results here in perspective.
> >
> > We have included NVAE and IAF in the CIFAR table.
> > We will also include the parameter counts for these bigger optimized models.

---

> > > ### Comment · AnonReviewer1 · 2020-11-21
> > > **Thanks for your response; unfortunately it does not resolve the most important questions.**
> > >
> > >
> > > Please see below, most importantly the first point.
> > >
> > > 1. First there was a mistake in my original review (and perhaps your response as well): in the stochastic estimator of the ELBO, the latents are sampled from $q$, so it is the bottom-level latent $z_1$ that follows a Gaussian.
> > > However, in any case the text doesn't seem right: the text defines $U_\rho\mu_i(z_{i+1})$ as $E_{z_\omega}(\mu_i(\rho z_{i+1} + \sqrt{1-\rho^2} z_\omega))$. It then says that the aglorithm works by replacing $N(z_i|\mu_i(z_{i+1}), \sigma_i(z_{i+1}))$ with $N(z_i|U_\rho \mu_i(z_{i+1}), U_\rho \sigma_i(z_{i+1}))$, and the parameter of the new latent distribution "have lower variance but the same expected value relative to the conditional input latent distribution". In your rebuttal you stated that this is valid because "in the analysis we only work with conditional distributions" (which are Gaussian).
> > > The problem is you can't justify the use of $\mu_i(z_{i+1})$ by arguing $q(z_{i+1}|z_i)$ (or any conditional distribution) is Gaussian. For example,
> > > you can't argue that $E(U_\rho \mu(z_{i+1})) = E(E(U_\rho \mu(z_{i+1}) | z_i)) \text{``}=\text{''} E(E(\mu(z_{i+1}) | z_i)) = E(\mu(z_{i+1}))$, since the second equality doesn't hold: to see this, let $\rho=0,\mu(x)\equiv x$ and $E(z_{i+1}|z_i)=1$, then LHS$=0\ne 1=$RHS.
> > > If you want to utilize the Gaussianity of the *conditional* distribution $q(z_{i+1}|z_i)$, or in other words, use the fact that the conditional expectation $E(\mu_i(z_{i+1})|z_i)$ is a function of *some* Gaussian random variable, you should replace $U_\rho \mu_i(z_{i+1})$ with
> > > $E_{z_\omega}\left(\mu_i\left(s_{i+1}\left(\rho \frac{z_{i+1} - m_{i+1}}{s_{i+1}} + \sqrt{1-\rho^2}z_\omega\right) + m_{i+1}\right)\right),$
> > > where $m_{i+1},s^2_{i+1}$ are the mean and variance of the conditional distribution, respectively.
> > > Is this what you are implementing? If not, please provide *complete* statements and proofs of generalized versions of Prop.1 and Prop.2 that support your algorithm.
> > >
> > > 2. When I refer to unbounded likelihood I do not mean that at some point of training you will see literally infinite likelihood (or ELBO), or zero output variance. What I mean is that the output variance will be approaching zero, and consequently the scale of the ELBO will be growing. If you lower bound the output variance by a small positive threshold, the resulted ELBO will still be quite large, which will affect training. Your newly added Figure 2 is consistent with the possibility that the minimum output variance (after smoothing to remove oscillation) is *approaching* zero: at the end of the curve it apparently has a negative slope, and the validation ELBO does not provide any evidence that training should be stopped.
> > > Also, while the unbounded likelihood issue is described under the setup of continuous likelihood, in your setup the conditional distributions $p(z_i|z_{i+1})$ are Gaussian, so it is reasonable to ask whether a similar issue will occur (by taking derivative on the ELBO you can see that the optimal variance of $p(z_i|z_{i+1})$ is the average reconstruction error of $z_i$, which could be small when the data distribution is complex and the model is good), even though alternative, VQ-VAE-like models exist. It is for this reason I asked for visualizations of the output variances.
> > >
> > > 3. "reducing variance at the cost of bias" isn't interesting on its own, since before you derive the coefficients of the leading terms in the big-O bounds, it is possible that mean squared error never improves (e.g. variance is $2\rho^2$ and squared bias is $1-\rho^2$). Of course the experiments show otherwise so this is not the most important concern, but experiments don't make an incomplete theory interesting.

---

> > > > ### Author Response · Authors · 2020-11-22
> > > > **Further Clarification (part 2)**
> > > >
> > > > >"reducing variance at the cost of bias" isn't interesting on its own, since before you derive the coefficients of the leading terms in the big-O bounds, it is possible that mean squared error never improves (e.g. variance is
> > > > and squared bias is
> > > > ). Of course the experiments show otherwise so this is not the most important concern, but experiments don't make an incomplete theory interesting.
> > > >
> > > >
> > > > To answer your concern, we show next that the leading coefficients of $\rho^2$ in the big O term are the same in both the bias and variance expressions.
> > > >
> > > > The expression for bias we derive is given by
> > > > $$\text{bias} = \frac{1}{2\sigma_p^2}(Var(\mu_p) - Var(U_\rho \mu_p)),$$ as in section 3 of the paper.
> > > > We want to compare the leading coefficients of the bias and the variance.
> > > > Let for clarity of exposition assume that the variance function of the VAE latent layer is constant, that is $\sigma(z_{i+1})=\sigma$, as in the analysis in section 3.
> > > > Then, we can rewrite the bias as
> > > > $$\text{bias} = (Var(\frac{\mu_p}{\sqrt{2}\sigma_p}) - Var(\frac{U_\rho \mu_p}{\sqrt{2}\sigma_p} )).$$
> > > > The variance term expands as
> > > > $$Var[U_\rho f]  =\sum_{\alpha: |\alpha| > 0} \rho^{2|\alpha|} \hat{f}(\alpha)^2.$$
> > > >
> > > > The leading term of the variance $Var(\frac{\mu_p}{\sqrt{2}\sigma_p})$ is the sum of squares of degree-1 coefficients.
> > > > If this leading term is equal to $c$, the respective leading term for the variance of the OU version with $\rho^2$ is also $c$ since the OU semigroup only multiples the first-order terms by $\rho$.
> > > > As a result, subtracting the two as per the formulation of bias above, the leading term of the bias is $c-c\rho^2=c(1-\rho^2)$. That is, the leading term of the bias is scaled by the same coefficient as the leading term of the variance.
> > > > As both the variance and the bias have the same leading rate $c$ with $\rho$, we cannot have the kind of difference suggested by the reviewer. A similar argument can be made for non-constant variance functions.
> > > >
> > > > Despite the above, the goal of the analysis is only to show the effect of smoothing operation on the KL divergence term, which is related to posterior collapse. The intent is not to show a provable improvement in the "mean squared error" solely by analytical means.
> > > >
> > > > We hope we addressed all remaining concerns. If not, please let us know. In any case, thank you for helping us clarifying the manuscript.

---

> > > > ### Author Response · Authors · 2020-11-22
> > > > **Further Clarification (part 1)**
> > > >
> > > >
> > > > >First there was a mistake in my original review (and perhaps your response as well): in the stochastic estimator of the ELBO, the latents are sampled from , so it is the bottom-level latent
> > > > that follows a Gaussian.
> > > > However, in any case the text doesn't seem right: the text defines $U_\rho \mu (z_{i+1})$ as $E_{z_\omega}[\mu(\rho z + \sqrt{1-\rho^2} z_\omega)]$ ...
> > > >
> > > >
> > > > Thank you for your errata. The main question is whether the noise operator $U_\rho$ preserves expectations in general Gaussian distributions. To shed light, we revisit some of the proposed statements in our method.
> > > >
> > > > 1. The OU semigroup $U_{\rho}$ is defined as $U_\rho[f(z)]=E_{z_\omega} [f(\rho z + \sqrt{1-\rho^2} z_\omega)]$, where $z,z_\omega \in N(0, 1)$ is a standard gaussian. The operator is defined for functions $f$ with standard gaussian input $z \in N(0, 1)$ in equation 3 in the original draft.
> > > >
> > > > 2. For a general Gaussian $z$ we reparameterize: $z=g(\epsilon) = \sigma\epsilon + \mu$. The function $f(g(\epsilon))$ is then a function on  a standard Gaussian $\epsilon \in N(0,1)$, on which the noise operator $U_\rho$ can be applied. The case for reparameterized $z$ was given in the original draft in equation 4 and the preceding text.
> > > >
> > > > 3. We take the expectation of the OU operated function and show that $E[U_\rho f(z)] = E[f(z)]$. Letting $z=g(\epsilon)=\sigma\epsilon + \mu$ we have $E[U_\rho f(z)]=E[U_\rho f(g(\epsilon))]=E[f(g(\epsilon))] = E[f(z)]$, where the second equality follows from proposition 2.
> > > >
> > > > For clarity of the explanation, we revisit the counterexample provided
> > > >
> > > > >$E(U_\rho \mu(z_{i+1}))=E(E(U_\rho \mu (z_{i+1} | z_i))) ``=''E(E(\mu(z_i+1)|z_i))=E(\mu(z_{i+1}))$.
> > > >
> > > >
> > > > Let $\rho = 0$ and let $E_{z_{i+1}|z_i}[z_{i+1}] = 1$, $\mu(x)=x$.
> > > > The left hand side
> > > > $$E_{z_{i+1}|z_i}[U_\rho \mu(z_{i+1})] = E_{z_{i+1}|z_i}E_\epsilon[\rho z_{i+1} + \sqrt{1-\rho^2}\epsilon]
> > > >   = E_{z_{i+1}|z_i}[0] = 0$$
> > > >
> > > > According to the proposed counterexample this is not equal to the right hand side $E_{z_{i+1}|z_i}[\mu(z_{i+1})] = E_{z_{i+1}|z_i}[z_{i+1}] = 1$.
> > > >
> > > > The counterexample is incorrect because it does not apply $U_\rho$ to the  standard normal Gaussian in reparameterized variable $z_{i+1}$ but on the variable $z_{i+1}$ itself.
> > > > To show that $E_{z_{i+1}|z_i}[z_{i+1}] = E_{z_{i+1}|z_i}[U_\rho z_{i+1}]$ we would correct the counterexample as follows.
> > > >
> > > > Let $z_{i+1} = \sigma \epsilon_{i+1} + \mu$. Then
> > > > $E_{z_{i+1}|z_i} U_\rho \mu(z_{i+1}) = E_{z_{i+1}|z_i} U_\rho (z_{i+1}) =  E_{z_{i+1}|z_i} E_\epsilon[\sigma(\rho \epsilon_{i+1} + \sqrt{1-\rho^2}\epsilon) + \mu] = E_{z_{i+1}|z_i}[\sigma\epsilon + \mu] = 1$
> > > >
> > > >
> > > >
> > > > >When I refer to unbounded likelihood I do not mean that at some point of training you will see literally infinite likelihood (or ELBO), or zero output variance. What I mean is that the output variance will be approaching zero, and consequently the scale of the ELBO will be growing. If you lower bound the output variance by a small positive threshold, the resulted ELBO will still be quite large, which will affect training ...
> > > >
> > > >
> > > > The reason unbounded likelihoods are undesirable is because they make bad generalizations (see proposition 1 in Mattei and Frellsen, 2018). Also see in, for example, figure 2 in the cited paper (Mattei and Frellsen, 2018) a case where output variance is left unconstrained and the training ELBO keeps improving but test ELBO is exceedingly poor. The fact that in our experiment with a Gaussian decoder, even in the unconstrained variance setting, in A.6 (figure 2) the train and validation ELBO are close already implies that the model has not chosen these bad optima.
> > > >
> > > > All the ELBO numbers in our paper are on either the validation or test set. The fact that the numbers are reasonable already implies that these bad optima were not found by the models.
> > > >
> > > > Furthermore, as we described before, all of this is not relevant for the analysis since it does not deal with the decoder output. The analysis only deals with intermediate latent distributions $p(z_i|z_{i+1})$ which only appear in KL divergence terms and obviously VAEs are not trying to maximize KL divergences so the problem of unbounded maximum likelihood is not relevant here.

---

> ### Author Response · Authors · 2020-11-21
> **We hope concerns have been met**
>
> We hope we have addressed all remaining concerns and you would consider raising the score. If not, please let us know of any further evidence you would like us to provide.

---

### Official Review · AnonReviewer3 · 2020-10-27
**Good theoretical contribution, however the experiments are not sufficient**

**Rating:** 4
**Confidence:** 4

**Review:**

1. Summary
This paper studies the training of deep hierarchical VAEs and focuses on the problem of posterior collapse. It is argued that reducing the variance of the gradient estimate may help to overcome posterior collapse. The authors focus on reducing the variance of the functions parameterizing the variational distribution of each layer using a layer-wise smoothing operator based on the Ornstein-Uhlenbeck semigroup (parameterized by a parameter $\rho$). The operator requires additional Monte-Carlo samples. The authors provide an analytical analysis of bias and variance. Last they train multiple VAEs models, measure the posterior collapse and observe a phase transition behaviour depending on the parameter $\rho$.

2. a Strong Points
This paper introduced a theoretically grounded solution to the problem of posterior collapse. In particular, it is discussed that variance may be an issue. Great efforts were invested to study the behaviour of the Hermite VAE in theoretical terms and the authors provide analytical results on the Bias and Variance for this estimator.

2. b Weak Points
* Complexity
In the main text, it is written that "*experiments show that 5 or 10 samples suffice*". This is a major drawback for Hermite VAEs and the complexity of the algorithm is not discussed, nor it is studied empirically. Given 5 MC samples, I interpret that HVAE is 5 times more expensive than other approaches -- please clarify this point.
* Empirical study of the variance
The problem of the variance is discussed in the paper but left apart in the experimental section. I would expect the authors to measure the variance (and/or SNR) of the gradients for the HVAE objective, the VAE, for advanced estimators such as STL and DReG. A study is required to corroborate the claim that reducing variance overcomes posterior collapse.
* Experiments on posterior collapse
I am surprised to see that none of the existing methods (KL warmup and freebits) allows overcoming posterior collapse (Figure 1). At least using the right amount of freebits should improve the results (the number of freebits is not reported). Furthermore, the authors should report the KL divergence in the benchmark experiment.
* Experimental protocol
I don't understand why VAE models trained in section 5 only have 2 layers whereas HVAE uses 4 layers: this is not a fair comparison. Furthermore, LVAE should be studied on the basis of posterior collapse -- not only in terms of likelihood.

3. Recommendation
Unfortunately, based on the current form of the paper, I recommend rejecting this paper.

4. Recommendation Arguments
Despite the good theoretical contributions, I do not find the experimental section to be strong enough to support the claims. In particular, the cost induced by the additional MC samples is not discussed and methods are hence not compared on the same basis.

5. Questions to the Author
- What is the complexity of HVAE? Do the VAE models use multiple MC samples as well?
- Why using only 2 layers for the VAE models?
- How are the freebits and KL-warmup applied in figure 1?

6. Feedback
Your work is very relevant and the theoretical insights are very interesting, this work would greatly benefit from an improved experimental section.

In the first page, two typos:
-  you defined $q(z | x)$ and not $q(x, z)$
- The KL divergence in equation 1 should depend on $q(z_i | z_{i-1})$ and $p(z_i | z_{i+1})$

---

> ### Author Response · Authors · 2020-11-19
> **Response (part 1)**
>
> Thank you for the review. We try to address the concerns below.
>
> >Complexity In the main text, it is written that "experiments show that 5 or 10 samples suffice". This is a major drawback for Hermite VAEs and the complexity of the algorithm is not discussed, nor it is studied empirically. Given 5 MC samples, I interpret that HVAE is 5 times more expensive than other approaches -- please clarify this point.
>
> We do provide a timing comparison in the appendix (A.3) for a 4 layer model on static MNIST. Comparisons show that our method bears a small only a extra cost in time per epoch (4.6 sec for our model, 4.2 sec for VAE, 4.4 sec for IWAE). The complexity of the method is approximately on-par with IWAE, which actually has more parallel computations than us. We will add a reference in the main paper.
>
> The reason for the smaller added complexity is that the OU sampling operation is only applied in chosen parts of the model. In particular, we do not apply the sampling operation in the inference network. Secondly, we do not apply the operation in the last stochastic layer of the decoder. The overall complexity (for 5 MC samples) is significantly less than if we were to repeat 5 samples with the original complexity.
>
> > Empirical study of the variance The problem of the variance is discussed in the paper but left apart in the experimental section. I would expect the authors to measure the variance (and/or SNR) of the gradients for the HVAE objective, the VAE, for advanced estimators such as STL and DReG. A study is required to corroborate the claim that reducing variance overcomes posterior collapse.
>
> We have added a gradient variance comparison between VAE and HVAE in the Appendix A.5. Experiments indeed show that variance is reduced with our Hermite VAEs.
>
> However, there are a few points which need to be emphasized about our method.
>
> First, since we have a variance-bias trade-off we cannot conclusively say variance reduction (at least on its own) is responsible for elimination of posterior collapse. We do claim, however, that OU smoothing, which also reduces variance, does mitigate posterior collapse. We will clarify this in the introduction.
>
> Furthermore, since the smoothing operator introduces a bias-variance trade-off, a direct comparison of variance with other unbiased methods is not meaningful. Also, it is not sufficient to establish the requisite link between smoothing and collapse.
>
> Last, we include a study on the relation between smoothing (and variance reduction thereafter) and posterior collapse in figure 2. The parameter $\rho$ controls the amount of smoothing (and variance reduction). By increasing $\rho$ (thus less smoothing, more variance) leads to posterior collapse. Taken together with the theory, this study establishes a link between smoothing and posterior collapse.
>
> > Experiments on posterior collapse I am surprised to see that none of the existing methods (KL warmup and freebits) allows overcoming posterior collapse (Figure 1). At least using the right amount of freebits should improve the results (the number of freebits is not reported). Furthermore, the authors should report the KL divergence in the benchmark experiment.
>
> We disagree with the conclusion. KL annealing and free bits do help in mitigating posterior collapse for shallow hierarchies of stochastic variables, annealing being more effective. For instance, in tables 1 and 2 (updated version) KL annealing and free bits do lead to more active units, especially in the lower levels, and a somewhat larger top layer KL divergence than a standard VAE. However these techniques are not very effective at overcoming collapse in deeper hierarchies, which is the motivation of our work.
>
> [We assume the reference here is to table 1 and not figure 1 where we plot KLD and validation ELBO curves for the vanilla VAE.]
>
> >Experimental protocol I don't understand why VAE models trained in section 5 only have 2 layers whereas HVAE uses 4 layers: this is not a fair comparison. Furthermore, LVAE should be studied on the basis of posterior collapse -- not only in terms of likelihood.
>
> The VAE models showed in section 5 in figures 1 & 2 and tables 1 & 2 (updated version) all were trained with 4 stochastic layers. The only place we show a 2-layer VAE is in table 4 where we compare against models reported in the literature. We were unable to find a case of a vanilla VAE deeper than 2 layers with better performance on this dataset in the literature. We will clarify this in the text. We have now also added a comparison to the 5 layer LVAE  in table 5. Hermite VAE outperforms all baselines.

---

> > ### Author Response · Authors · 2020-11-19
> > **Response (part 2)**
> >
> > >Questions to the Author
> > >
> > > What is the complexity of HVAE? Do the VAE models use multiple MC samples as well?
> >
> > Please see the first point about the complexity of HVAE. In short, it is on-par with IWAE. Among the other models, the IWAE models use a multi sample objective and we used 5 samples for IWAE in tables 1 and 2.
> >
> > >Why using only 2 layers for the VAE models?
> >
> > Please see the response to the "Experimental Protocol" point above.
> >
> > >How are the freebits and KL-warmup applied in figure 1?
> >
> > KL-warmup is applied by setting the KL coefficient to 0 for the first 10,000 steps. The coefficient is linearly increased to 1 over the next 500,000 steps.
> >
> > For free bits we use the recommended settings as in the IAF paper. We apply the free bits separately per layer. We experimented with free bit values in {0.5,1.0,2.0,3.0}. Larger values caused the model to become unstable. The models in table 2 (updated version) use a value of 1.0. The models in the new table 1 use a value of 2.0. In most cases with higher free bit values we found the training to slow down significantly. This is the reason for the lower ELBO values for the free bits models in tables 1 and 2.
> > The models in these tables were trained for 1 million time steps since posterior collapse manifests itself relatively early in training. It is likely that ELBO values would improve with longer training time with free bits. The experiments section has been updated with the details of KL warmup and free bits
> >
> > >Feedback Your work is very relevant and the theoretical insights are very interesting, this work would greatly benefit from an improved experimental section.
> >
> > We thank the reviewer for the kind comments and hope that the concerns have been met.

---

> > > ### Comment · AnonReviewer3 · 2020-11-20
> > > **Great improvements in the empirical section however some issues remain (complexity)**
> > >
> > > The experimental section was greatly improved in this revision. Thank you for answering my concerns, I now have a clearer picture.
> > >
> > > > Complexity In the main text, it is written that "experiments show that 5 or 10 samples suffice". This is a major drawback for Hermite VAEs and the complexity of the algorithm is not discussed, nor it is studied empirically. Given 5 MC samples, I interpret that HVAE is 5 times more expensive than other approaches -- please clarify this point.
> > >
> > > The added wallclock time does not answer my concern because GPU memory is the bottleneck in your case. If I understand correctly, a VAE requires M samples while the IWAE and HVAE require M x K samples, which probably prevent training deep VAEs (table 3 is limited to 6 layers max). In other words, please correct me if I am wrong, HVAE use x5 GPU memory. If this is the case, HVAE is not directly comparable with other methods and this should be explicit.
> > >
> > > > Empirical study of the variance The problem of the variance is discussed in the paper but left apart in the experimental section. I would expect the authors to measure the variance (and/or SNR) of the gradients for the HVAE objective, the VAE, for advanced estimators such as STL and DReG. A study is required to corroborate the claim that reducing variance overcomes posterior collapse.
> > >
> > > Thank you for the detailed explanation, this is much clearer.
> > >
> > > > Experiments on posterior collapse I am surprised to see that none of the existing methods (KL warmup and freebits) allows overcoming posterior collapse (Figure 1). At least using the right amount of freebits should improve the results (the number of freebits is not reported). Furthermore, the authors should report the KL divergence in the benchmark experiment.
> > >
> > > I was indeed misinterpreting the posterior collapse experiment. The tables 1 and 2 provide a clear picture. HVAE appear to be effective at mitigating posterior collapse.
> > >
> > >
> > > ## Review update:
> > >
> > > Overall, the revision answers many of my original concerns (study of the posterior collapse, variance). There is now a good connection between the theory and the different experiments, but the benchmark experiments remain unsatisfactory. I would be willing to rise my score, although this paper remains, in my opinion, on the grey line for acceptance.
> > >
> > > Computational complexity remains the main issue: M samples for VAEs vs. MxK samples for IWAE/HVAE. This means HVAE requires K times more memory than VAEs, which is costly and prevents using very deep VAEs. Hence, the comparison between HVAE and LVAE+ incomplete because only the number of parameters is reported. Yet the number of parameters is not the bottleneck in modern deep learning, GPU memory is. In the best-case scenario, you should demonstrate that deeper HVAEs are competitive with SOTA VAEs (BIVA, NVAE, IAF), although SOTA is not necessary and I believe a fair comparison is sufficient.

---

> > > > ### Author Response · Authors · 2020-11-21
> > > > **Clarification of memory consumption**
> > > >
> > > > > Computational complexity remains the main issue: M samples for VAEs vs. MxK samples for IWAE/HVAE. This means HVAE requires K times more memory than VAEs, which is costly and prevents using very deep VAEs. Hence, the comparison between HVAE and LVAE+ incomplete because only the number of parameters is reported. Yet the number of parameters is not the bottleneck in modern deep learning, GPU memory is. In the best-case scenario, you should demonstrate that deeper HVAEs are competitive with SOTA VAEs (BIVA, NVAE, IAF), although SOTA is not necessary and I believe a fair comparison is sufficient.
> > > >
> > > > Thank you for clarifying your concern regarding memory complexity. The proposed OU sampling is applied in selected layers only. In particular it is NOT applied
> > > >
> > > > 1. anywhere in the inference network, so the memory usage for the inference network is exactly the same as for a standard VAE.
> > > > 2. in the final stochastic layer of the decoder, i.e., the layer computing $p(x|z_1)$, so the memory usage is again the same as for a standard VAE.
> > > >
> > > > More generally, the OU sampling is applied *only in the decoder* network and *only for* sampling between latent layers *$p(z_i | z_{i-1})$*.
> > > >
> > > > The fact that OU sampling is not applied in the final stochastic layer $p(x|z_1)$ is especially important for deep VAEs, since the last decoder layer is also the largest and heaviest one so that to match image dimensions.  Thus in our proposal with 5 samples we additionally require 5 times the memory requirement for $p(z_2|z_1)$, $p(z_3|z_2)$, $p(z_4|z_3)$ only for a 4 stochastic layer model. When taken together with the fact that in many architectures higher layers are typically of smaller dimension, the total memory required is significantly less than 5 times the memory requirement of the basic VAE model. To quantitatively support our justification, we report the memory usage 4 and 6-layer models on CIFAR-10, using 5 deterministic layers per stochastic layer with 100 units per layer and 5 OU or IWAE samples. The memory usage is the maximum amount of used memory with a batch size of 64. For LVAE and BIVA we use code from the BIVA authors' pytorch repository. HVAE stands for the proposed model with OU sampling.
> > > >
> > > > |Model | Layers| Memory| #Parameters (M) |
> > > > |----- |:-----:|:-----:|:-----:|
> > > > |VAE   | 4 | 2.12G |  12.4  |
> > > > |VAE   | 6 | 2.45G |  16.8  |
> > > > |IWAE  | 4 | 8.59G |  12.4  |
> > > > |IWAE   | 6 | 10.5G |  16.8  |
> > > > |LVAE  | 15 | 6.4G  | 59.5  |
> > > > |BIVA  | 15 | 10.3G | 103  |
> > > > |HVAE  | 4 | 3.74G |  12.4  |
> > > > |HVAE   | 6 | 4.6G |  16.8  |
> > > >
> > > > We conclude that the additional memory cost incurred by the OU sampling is small and the final models are considerably smaller than IWAE, LVAE and BIVA.
> > > >
> > > > > Overall, the revision answers many of my original concerns (study of the posterior collapse, variance). There is now a good connection between the theory and the different experiments, but the benchmark experiments remain unsatisfactory. I would be willing to rise my score, although this paper remains, in my opinion, on the grey line for acceptance.
> > > >
> > > > Regarding benchmarks, we note that our primary focus is on *posterior collapse*, a fundamental problem when stacking stochastic variables. To this, we present a theoretically sound -and dare say fundamental- argument, as agreed by all reviewers, on the basis of analysis of functions on Gaussian spaces. Our theory and solution is experimentally confirmed under *all* settings and multiple datasets before and after the reviews, and uses only a basic architecture which establishes further the effectiveness of the approach. With these simple architectures we can already attain competitive likelihoods (3.39 BPD on CIFAR-10) on the standardized benchmarks and we expect our method to be orthogonal to other, architecture-driven SoTA approaches like NVAE. We believe that optimizing the architecture to clearly surpass SoTA requires its own separate study and should be considered beyond the scope of our current proposal.
> > > >
> > > > We hope we have addressed all remaining concerns and you would consider raising the score. If not, please let us know of any further evidence you would like us to provide.

---

> > > > > ### Comment · AnonReviewer3 · 2020-11-23
> > > > > **My main concern has been answered**
> > > > >
> > > > > Thank you for your detailed answer.
> > > > >
> > > > > I agree that pursuing SOTA is not necessary for this work and the experimental section demonstrates the ability of HVAE to overcome posterior collapse.
> > > > >
> > > > > Given that you update the paper with a paragraph detailing the complexity and memory consumption (as presented in the rebuttal), I am willing to update my score to 6.

---

> > > > > > ### Author Response · Authors · 2020-11-23
> > > > > > **Draft updated with complexity details**
> > > > > >
> > > > > > Thank you for your response.
> > > > > >
> > > > > > We have updated the paper with a paragraph describing the computational and memory complexity in section 2 on page 4. The table comparing the empirical memory of various models has been added as table 6 on page 9.
> > > > > >
> > > > > > We hope that this addition satisfies your requirement and thank you again for your review.

---

### Official Review · AnonReviewer2 · 2020-10-31
**Good theoretical analysis; doubts on practical performance**

**Rating:** 6
**Confidence:** 3

**Review:**

This paper proposes a Hermite variational auto-encoder which use Ornstein Uhlenbeck Semi-group to p(z_i|z_i+1) which i denotes the latent layer number. It has clear theoretical inspiration and had solid analysis on variance reduction.

Pros:
Quality: The paper's generic theoretical motivation and analysis is with high quality.
Clarify: The paper's presentation is clear.
Originality: This paper provides a new perspective and used mathematically tools of Hermite expansion etc to inspire and proposed new method for variance reduction which prevent dying unity problem in Hierarchical VAE. Although motived by advanced tools, but the application of the method in vanilla version of hierarchical VAE (in term of implementation) seems very simple. Thus, the method looks easy to adopt.

Cons and questions:
Significance:
1) Does the method work for vanilla VAE with only one level of z? It seems that it is only applicable to the hierarchical version as the operator is applied in  p(z_i|z_i+1) and if it is one level, it lost the point due to single Normal prior.  This may limit the application impact as VAE is much more widely adapted in different applications comparing to hierarchical VAE.
2) I am not sure making unit not dying at all is desired (such as shown in last column of table one or Figure 3. Being Bayesian with the prior, there is a natural model selection behavior (implicit Occam's Razor) , thus, behavor such as the method with active unints (40,40,40,24) in table one may not be desired and rather a bit weird as only the last layer have dying units. Behavior such as VAE+KL (2,3, 11, 37) looks more natural to me as simpler model is needed in high hierarchy.
3) Experiments only compared to Ladder VAE in number of dying unit but not in term of ELBO for performance. As LVAE is one of the most known work in this domain and also mentioned first in the related work section), this is weird. In LVAE paper, the MNIST performance is reported as -81-82 while in the paper it is about -85 which is significantly worse. Although there is a chance due to minior setting differences, I doubt the method's performance can match LVAE.  (Again with 2), I don't think that puring comparing the number of units without reporting performance makes sense).
4) in term of performance of ELBO, most of the time, it does not match simple KL annealing either.
5) there are more highly related work anaylsis the variance-bias trade off such as Tighter Variational Bounds are Not Necessarily Better are not discussed in the paper.

---

> ### Author Response · Authors · 2020-11-19
> **Response (part 1)**
>
> We thank the reviewer for appreciating the contribution. We hope to address the concerns below.
>
> >Cons and questions: Significance:
> >Does the method work for vanilla VAE with only one level of z? It seems that it is only applicable to the hierarchical version as the operator is applied in p(z_i|z_i+1) and if it is one level, it lost the point due to single Normal prior. This may limit the application impact as VAE is much more widely adapted in different applications comparing to hierarchical VAE.
>
> Posterior collapse is most clearly observed in bigger (hierarchical) stochastic depths in VAEs, with strong autoregressive decoders. Or, as Figure 3 in the paper shows, in shallower but wider VAE architectures.  Existing methods for reducing collapse already work quite well with shallow models (as long as the decoder is not too powerful). Although the focus of our paper in posterior collapse with increasing stochastic depth, the proposed model works just as well for a single stochastic layer. We show this in Table 2 in A.2 in the Appendix, where Hermite VAE improves latent activity (compared to regular VAE) on single stochastic layer architecture on MNIST (18 active units for VAE v. 31 for HVAE out of 64).
>
> We conclude that the proposed Hermite VAE can also work with single stochastic layers.
>
> >I am not sure making unit not dying at all is desired (such as shown in last column of table one or Figure 3. Being Bayesian with the prior, there is a natural model selection behavior (implicit Occam's Razor) , thus, behavor such as the method with active unints (40,40,40,24) in table one may not be desired and rather a bit weird as only the last layer have dying units. Behavior such as VAE+KL (2,3, 11, 37) looks more natural to me as simpler model is needed in high hierarchy.
>
> We test the hypothesis of whether posterior collapse does some sort of feature selection pruning in higher layers as follows.
> We train multiple 4-layer VAEs+KL annealing with latent dimensions: {40,40,40,40}, {30,30,30,30}, {20,20,20,20}, {10,10,10,10}, {5,5,5,5}. If posterior collapse is a means for feature selection pruning, we would expect a good balance between (A) the validation ELBO while (B) attaining larger relative top layer activity (active/total units) for smaller networks.
>
> The results are (see Appendix A.4 for details):
>
> |Model  | V. ELBO  |  Active units | Relative activity |
> |:------------|:---------------------:|:-------------------:|:-------------------:|
> |40-40-40-40 | -84.73     | 1,6,15,40  | 2.5%, 15%, 37.5%, 100% |
> |30-30-30-30 | -88.7       |1,6,13,30  | 2.5%, 15%, 32.5%, 75% |
> |20-20-20-20 | -85.4       | 1,5,12,20  | 2.5%, 12.5%, 30%, 50% |
> |10-10-10-10 | -90.47     | 2,3,9,10   | 5%, 7.5%, 22.5%, 25% |
> |5-5-5-5         |-104.36    | 1,3,5,5 |     2.5%, 7.5%, 12.5%, 12.5% |
>
> In summary, with smaller latent dimension the validation ELBO becomes worse. At the same time, the relative top layer activity remains the same. If the model was performing feature selection, we would expect not to have worse validation ELBO alongside low top layer activity. Especially for the smaller models.
>
> We conclude that, according to experiments, there is no sign that posterior collapse is the way of the model to do feature selection pruning in the higher layers.

---

> > ### Author Response · Authors · 2020-11-19
> > **Response (part 2)**
> >
> > > Experiments only compared to Ladder VAE in number of dying unit but not in term of ELBO for performance. As LVAE is one of the most known work in this domain and also mentioned first in the related work section), this is weird. In LVAE paper, the MNIST performance is reported as -81-82 while in the paper it is about -85 which is significantly worse. Although there is a chance due to minior setting differences, I doubt the method's performance can match LVAE. (Again with 2), I don't think that puring comparing the number of units without reporting performance makes sense).
> > >
> > > in term of performance of ELBO, most of the time, it does not match simple KL annealing either.
> >
> > We apologize for the misunderstanding. The intent of that table was to show we obtain reasonable ELBO and layer activity in the simple MNIST case. The LVAE paper reported numbers on dynamic MNIST, while we originally reported scores on the static MNIST. This has a big impact on likelihood values (see Tomczak and Welling, 2017, Table 2 and 3). Also, results on static MNIST, when comparing posterior collapse in table 1 & 2 (in the update), were cut short after only 1M steps for computational reasons. When repeating the experiment on dynamic MNIST, with the same latent dimensions as in LVAE, and full optimization, then we obtain the following results (see table 5 in updated paper):
> >
> > |Model |Test ELBO|
> > |:---------|:------------:|
> > |HVAE |-81.1|
> > |LVAE |-81.7|
> >
> > For the Hermite VAE we use the same number of latent dimensions like in LVAE: 64-32-16-8-4 (5 stochastic layers). We observe that on the same data we outperform LVAE. We clarify in Table 5 in the paper.
> >
> > After your suggestion, we repeated experiments on all collapse mitigation methods (including KL annealing) on the 64-32-16-8 architecture on dynamic MNIST. We present results in Table 1 in the updated pdf. With the new architecture KL annealing worked a bit better and with more active units (also compared to the LVAE paper). Still, Hermite VAE attains better validation ELBO and has more active units, thus addressing posterior collapse better. To clarify the differences between tables, in the updated pdf:
> > - in Tables 1, 2 we include the comparisons for posterior collapse (dynamic MNIST)
> > - in Tables 4, 5 we include the optimized models with optimized $\rho$ parameter, test ELBO (instead of val ELBO) computed with 5,000 importance weighed samples (instead of 100).
> >
> > We conclude that on the same data and experimental setup, the proposed Hermite VAE obtains better ELBOs and has more active units on dynamic MNIST (Table 1), as well as the more complex and realistic CIFAR (Table 3), including LVAE and KL annealing.
> >
> > >there are more highly related work anaylsis the variance-bias trade off such as Tighter Variational Bounds are Not Necessarily Better are not discussed in the paper.
> >
> > We appreciate the reference suggestion and will include it in the update.

---

> ### Author Response · Authors · 2020-11-21
> **We hope concerns have been met**
>
> We hope we have addressed all remaining concerns and you would consider raising the score. If not, please let us know of any further evidence you would like us to provide.

---

### Author Response · Authors · 2020-11-19
**Summary of changes**

The reviewers agree on the usefulness of the theoretical contribution, but mostly have had concerns about the experimental results. Some of the concerns stemmed from a miscommunication where results on different datasets (static v. dynamic MNIST) were compared: We have added new results on dynamic MNIST to section 5 comparing posterior collapse and also a comparison with Ladder VAE in terms of test likelihood . Other concerns were about complexity where we clarify that  our method's empirical wall time is on par with IWAE while memory usage is significantly less than IWAE in the comments below and in the appendix.

The new additions in the paper are a paragraph on complexity with empirical comparison of memory usage, results on dynamic MNIST in the main paper. And experiments on output variance, 1-layer models and empirical verification of the hypothesis that the top layers might be pruning units rather than collapsing in the appendix.

---

### Decision · Program_Chairs · 2021-01-07
**Final Decision**

**Decision:**

Reject

**Comment:**

This paper develops a smoothing procedure to avoid the problem of posterior collapse in VAEs. The method is interesting and novel, the experiments are well executed, and the authors answered satisfactorily to most of the reviewers' concerns. However, there is one remaining issue that would require additional discussion. As identified by Reviewer 1, the analysis in Section 3 is only valid when the number of layers is 2. Above that value, "it is possible to construct models where the ELBO has a reasonable value, but the smoothed objective behaves catastrophically". Thus, the scope of the analysis in Section 3 deserves further discussion. Given the large number of ICLR submissions, this paper unfortunately does not meet the acceptance bar. That said, I encourage the authors to address this point and resubmit the paper to another (or the same) venue.